

# MSSA: multi-stage semantic-aware neural network for binary code similarity detection

Bangrui Wan[1,2], Jianjun Zhou[1], Ying Wang[1], Feng Chen[1,2] and Ying Qian[1,2]

[1] School of Software Engineering, Chongqing University of Posts and Telecommunications, Chongqing, China
[2] Chongqing Engineering Research Center of Software Quality Assurance, Testing and Assessment, Chongqing, China

## ABSTRACT

Binary code similarity detection (BCSD) aims to identify whether a pair of binary code snippets is similar, which is widely used for tasks such as malware analysis, patch analysis, and clone detection. Current state-of-the-art approaches are based on Transformer, which require substantial computation resources. Learning-based approaches remains room for optimization in learning the deeper semantics of binary code. In this paper, we propose MSSA, a multi-stage semantic-aware neural network for BCSD at the function level. It effectively integrates the semantic and structural information of assembly instructions within and between basic blocks, and across the entire function through four semantic-aware neural networks, achieving deep understanding of binary code semantics. MSSA is a lightweight model with only 0.38M parameters in its backbone network, suitable for deployment in CPU environments. Experimental results show that MSSA outperforms Gemini, Asm2Vec, SAFE, and jTrans in classification performance and ranks second only to the Transformer-based jTrans in retrieval performance.

## INTRODUCTION

Binary code similarity detection (BCSD) aims to identify whether a pair of binary code snippets is similar. It has broad application value across multiple fields, including malware analysis (*Cesare, Xiang & Zhou, 2013*; *Shirani et al., 2018*; *Xu et al., 2017a*; *Liu et al., 2018*), detection of software piracy (*Luo et al., 2014*; *Luo et al., 2017*), patch analysis (*Xu et al., 2017b*; *Kargén & Shahmehri, 2017*), reverse engineering (*Luo et al., 2023*), and vulnerability detection (*Gao et al., 2018*; *Eschweiler, Yakdan & Gerhards-Padilla, 2016*; *David, Partush & Yahav, 2018*). In these practical scenarios, software is often closed-source or its source code is difficult to access, highlighting the importance of BCSD. BCSD typically use functions as the basic unit of analysis.

Traditional BCSD approaches often rely heavily on specific features of binary code. Approaches like BinHunt (*Gao, Reiter & Song, 2008*) and BinDiff (*Zynamics, 2021*) extract syntactic information from functions by capturing control flow graphs (CFG) and then compute the similarity between two functions. Approaches like TEDEM (*Pewny et al., 2014*)

Corresponding author
Ying Qian, qianying@cqupt.edu.cn

and XMATCH (*Feng et al., 2017*) use graph or tree edit distances to evaluate code similarity. BinGold (*Alrabaee, Wang & Debbabi, 2016*) and Libv (*Qiu, Su & Ma, 2016*) detect binary code similarity from the semantic and CFG perspectives using graph algorithms. However, these approaches often struggle to capture the in-depth semantics of binary code and usually present unsatisfactory accuracy.

With the rapid advancement of machine learning technologies, learning-based BCSD approaches have emerged as a focal point of research, characterized by representing binary code as vector embeddings and calculating the similarity within the vector spaces. For instance, Asm2Vec (*Ding, Fung & Charland, 2019*), InnerEye (*Zuo et al., 2019*), and SAFE (*Massarelli et al., 2019a*) adopt deep neural networks to convert assembly instructions and functions into vector representations, capturing in-depth semantic information of the code. Genius (*Feng et al., 2016*), Gemini (*Xu et al., 2017a*), and Vulseeker (*Luo et al., 2014*) employ graph neural networks (GNN) to learn representations of functions' attributed control flow graphs (ACFG) and then compute their similarity. Recently, Transformer-based approaches such as jTrans (*Wang et al., 2022*) and Trex (*Pei et al., 2020*), leveraging the self-attention mechanism, grasp a thorough and comprehensive understanding of the contextual relationships within code sequences, demonstrating exceptional performance.

Despite the impressive progress, there still exist main challenges:

Firstly, Transformer-based BCSD approaches, which typically have a large number of parameters, require substantial computational resources and time for both training and execution. Consequently, training and deploying these models on devices with limited computational resources and time constraints is challenging.

Secondly, learning-based BCSD approaches often focus on either the semantic or structural information of binary code. However, the potential optimization from integrating both structural and semantic information is not yet fully explored.

Given these challenges, it is worthwhile in the BCSD field to investigate how to reduce dependency on computational resources while maintaining desirable performance. In this paper, we propose MSSA, a multi-stage semantic-aware neural network for BCSD with a backbone network of only 0.38M parameters. MSSA effectively integrates the semantic and structural information of assembly instructions within and between basic blocks, and across the entire function. This integration is implemented through four semantic-aware neural networks, namely Block Embedding, Intra-Block Learning, Inter-Block Learning, and Function-Level Learning. In doing so, MSSA would be able to grasp a deep understanding of binary code semantics. We evaluated the performance of MSSA against approaches including Gemini, Asm2Vec, SAFE, and jTrans on BinaryCorp-3M (*Wang et al., 2022*) dataset.

In summary, we have made the following contributions:

(1) We propose MSSA, a multi-stage semantic-aware neural network for BCSD. MSSA has a model size of only 0.38M parameters, enabling its deployment in CPU environments. The code for MSSA is available at GitHub (https://github.com/SQAbin/MSSA).

(2) We demonstrate that the multi-stage learning strategy employed by MSSA is crucial for enhancing BCSD. This approach effectively captures semantic and structural

information through Intra-Block Learning, Inter-Block Learning, and Function-Level Learning, significantly improving BCSD performance.

(3) We conducted extensive experiments to evaluate MSSA. The results demonstrate that MSSA outperforms Gemini, Asm2Vec, SAFE, and jTrans in classification performance, while ranking second only to the Transformer-based jTrans in retrieval performance.

## RELATED WORK

BCSD approaches can be categorized in three classes: traditional approaches, learning-based approaches, and Transformer-based approaches. The following section will detail related work pertaining to each of these categories.

### Traditional approaches

Traditional approaches for BCSD primarily fall into two categories: static and dynamic analyses. Static approaches utilize structured information such as CFG and Call Graphs (CG), along with graph/tree edit distance techniques, to assess binary code similarity. For instance, tools like BinClone (*Farhadi et al., 2014*), BinSign (*Nouh et al., 2017*), and BinShape (*Shirani, Wang & Debbabi, 2017*) leverage statistical features or graphs of CFG and CG for similarity analysis, whereas TEDEM (*Pewny et al., 2014*) and XMATCH (*Feng et al., 2017*) employ graph or tree edit distances. On the other hand, dynamic approaches rely on information collected during program execution including symbolic execution and deep taint analysis *etc.*, aiming to determine code similarity based on runtime behavior. Approaches such as iBinHunt (*Ming, Pan & Gao, 2012*) extract semantic information of functions through symbolic execution and deep taint analysis, while BinGo (*Chandramohan et al., 2016*) and BinGo-E (*Xue et al., 2019*) obtain function input/output values by executing target programs. HGE-BVHD (*Xing et al., 2024*) enhances BCSD accuracy by integrating basic block features into function representations using heterogeneous graph embeddings, which adapt to control and data flows for better discrimination of non-homologous functions.

In summary, traditional approaches have limitations in capturing the deep semantic information of binary code, especially in complex scenarios where compiler optimizations are applied. As a result, they usually do not offer satisfactory performance in BCSD. Some of these approaches do not rely on machine learning models, thus eliminating concerns about model size. Others that do employ machine learnings typically have smaller model sizes because they focus directly on analyzing binary code features or execution behaviors without incorporating complex model training processes. Traditional BCSD approaches struggle to handle compiler optimizations, leading to reduced detection accuracy. Additionally, these approaches have shallow semantic understanding, making it difficult to identify structurally different but semantically similar code. Furthermore, they rely on specific execution environments, resulting in poor scalability and subpar performance when dealing with complex scenarios and large-scale data.

### Learning-based approaches

In recent years, the rapid development of deep learning technology has introduced new approaches and techniques for BCSD. By constructing complex neural network models

such as Convolutional Neural Network (CNN), Recurrent Neural Network (RNN), and Graph Neural Network (GNN), these approaches can automatically learn and extract deep features of code to capture more profound semantic information, thereby enhancing the accuracy and efficiency of BCSD. For instance, $\alpha$diff (*Liu et al., 2018*) utilizes CNN to learn function embeddings directly from raw byte sequences, while VulSeeker extracts features from basic blocks and inputs them into a Deep Neural Network (DNN) to generate function embeddings for vulnerability function searches. Studies like Gemini (*Xu et al., 2017a*), Genius (*Feng et al., 2016*), and GraphEmb (*Massarelli et al., 2019b*) utilize ACFG and employ GNN to learn graph embeddings of the code, effectively capturing the structural and semantic information of programs. The successful application of Natural Language Processing (NLP) techniques in understanding textual semantics has also inspired new approaches in the field of BCSD. By treating binary code as a form of "language", researchers have attempted to use NLP models and techniques, such as word2vec and LSTM, to learn semantic representations of code. Studies like Asm2Vec (*Ding, Fung & Charland, 2019*), InnerEye (*Zuo et al., 2019*), and SAFE (*Massarelli et al., 2019a*) translate assembly instructions and functions into vector representations using NLP, capturing deep semantic information of binary code.

In summary, learning-based approaches can learn and capture the deep semantic information of binary code, increasing the accuracy of BCSD. However, existing approaches often focus on learning either the semantic or the structural aspect, and there is still room for improvement by simultaneously capturing both the semantic and structural information. Deep learning models require a large number of parameters to capture complex data features, typically ranging from tens of thousands to tens of millions. For example, the backbone network of SAFE has over 4.47M parameters. Additionally, these approaches rely on large-scale, high-quality training data and face challenges of limited generalizability of models.

## Transformer-based approaches

The approaches based on Transformer (*Vaswani et al., 2017*) demonstrate exceptional performance in BCSD. The self-attention mechanism of Transformer enables them to understand long-distance dependencies in instruction sequences and their corresponding dynamic values, making them adept at learning the subtle behaviors of functions. jTrans (*Wang et al., 2022*) is the first to incorporate the control flow information of binary code into a Bidirectional Encoder Representations from Transformers (BERT) (*Devlin et al., 2019*) model, employing novel jump-aware representations and customized pre-training tasks for BCSD. Trex (*Pei et al., 2020*) utilizes a new neural architecture called hierarchical Transformer, specifically designed to capture execution semantics from micro-traces during the pre-training phase. Codeformer (*Liu et al., 2023*), a model that nests GNN within a Transformer, leverages the strengths of both GNN and Transformer to effectively identify and compare similar segments within binary code. UniASM (*Gu, Shu & Hu, 2022*), another Transformer-based binary code embedding model, designs two new training tasks that make the spatial distribution of the generated vectors more uniform, allowing these vectors to be directly used for BCSD without any further fine-tuning.

In summary, Transformer-based approaches exhibit significant advantages in understanding the contextual information of binary code and in mining its deep semantic features. However, they typically have a very large number of parameters, often reaching hundreds of millions to billions. For example, the backbone network of jTrans has over 14.76M parameters, which is already a very small one based on the Transformer. These approaches far exceed traditional approaches and learning-based approaches in terms of resource consumption. Their high training and inference costs severely limit their application in resource-constrained environments.

Overall, as research approaches of BCSD have evolved, there has been a transition from traditional approaches to learning-based techniques, and now to transformer-based methods, all of which increasingly depend on computational resources. In the absence of costly computing power, advancing related research becomes a challenge. Existing studies primarily concentrate on improving model detection performance but often overlook some scenarios with limited computational resources. Moreover, the current trend of BCSD is to treat code as natural language. However, there is a lack of analysis from a code perspective. For example, for assembly instructions, each line of code initially has a strong semantic correlation with its adjacent context, then extending further to encompass interactions within and between code blocks, which may ultimately affect the entire function. This kind of multi-stage analysis from a code perspective is currently lacking in BCSD.

## DESIGN

### Overview

To ensure high detection performance while reducing dependency on computational resources, we propose MSSA, a multi-stage semantic-aware neural network for BCSD. MSSA effectively integrates the semantic and structural information of assembly instructions within and between basic blocks, and across the entire function through four semantic-aware neural networks. This integration enables a comprehensive and precise detection of similarities in binary code.

Figure 1 shows an overview of MSSA. MSSA comprises three main phases: Pre-Processing, Embedding Network, and Similarity Detection.

MSSA takes a pair of disassembled functions as input. Initially, the input undergoes the Pre-Processing phase, which includes the normalization of the disassembled functions and the construction of the functions' CFG and adjacency matrix as outputs.

Subsequently, the Embedding Network phase, which is also the core component of MSSA, encompasses four semantic-aware embedding networks in sequence: Block Embedding, Intra-Block Learning, Inter-Block Learning, and Function-Level Learning. They represent four stages of learning. The Embedding Network ultimately produce feature vectors that integrate deep semantic and structural information of disassembled functions.

Finally, the Similarity Detection component, built on top of the previous two components and based on the siamese neural network architecture and a Multi-Layer Perceptron (MLP) network, determines whether a given pair of functions is similar.

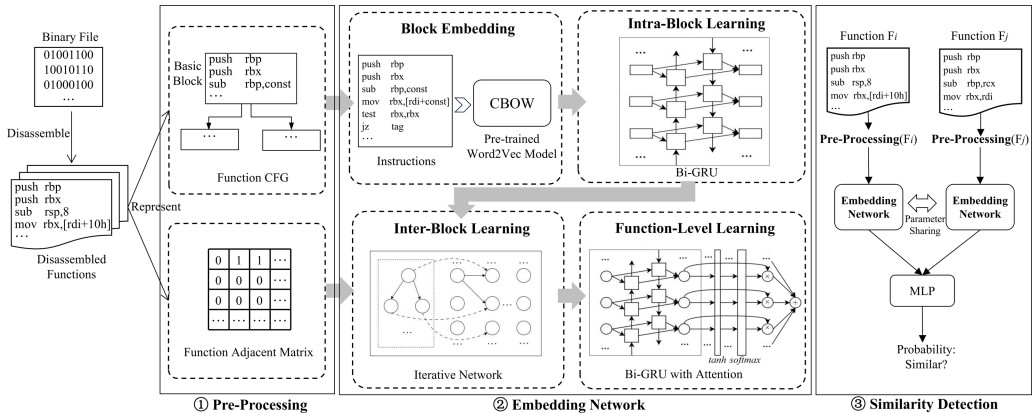

**Figure 1** The Overview of MSSA. MSSA contains three main phases Pre-processing, Embedding Network, and Similarity Detection, where the Embedding Network phase has four stages Block Embedding, Intra-Block Learning, Inter-Block Learning, and Function-Level Learning.

## Pre-processing

Like most BCSD approaches, we utilize the reverse engineering tool Interactive Disassembler Pro (IDA Pro) (*Hex-Rays, 2015*) to disassemble binary code files, deriving a set of disassembled functions. Each function is a sequence of assembly instructions. To enhance feature learning of assembly instructions, we incorporate a Pre-Processing phase, consisting of two tasks: Function Representation and Instruction Normalization.

## Function representation

For those disassembled functions containing conditional branching, jumping, looping, *etc.*, basic blocks can be further extracted. In MSSA, we represent a function by constructing its CFG at the basic block level. In the constructed CFG, nodes represent basic blocks, and edges represent control flow transitions during the execution of the function. Each basic block contains a sequence of assembly instructions.

Additionally, the CFG is also represented as an adjacency matrix. Nodes in the CFG are mapped to the rows and columns of the matrix respectively. Matrix elements are used to indicate whether there is a connecting edge between nodes. Given a control flow graph $G = (V, E)$, where $V$ is the set of basic block nodes and $E$ is the set of edges between basic blocks, in the corresponding adjacency matrix $A$, the matrix element for any two basic block nodes $V_i$ and $V_j$ in the graph $G$ is defined as follows:

$$A_{ij} = \begin{cases} 1, & (V_i, V_j) \in E \\ 0, & (V_i, V_j) \notin E \end{cases}, \tag{1}$$

where $A_{ij}$ represents the element in the $i$-th row and $j$-th column of the adjacency matrix, used to indicate whether there is an edge between $V_i$ and $V_j$. If there is an edge connecting these two nodes, $A_{ij}$ is 1; otherwise, it is 0.

*Instruction normalization*

In the field of BCSD, when encountering assembly instructions that were not included in the training data, referred to as the Out-of-Vocabulary (OOV) issue, it can significantly impair the detection performance. To address the OOV issue, it is helpful to abstract and standardize the opcodes and operands in assembly instructions. We apply the following strategies to normalize assembly instructions and reduce their vocabulary size:

(1) Keep all opcodes and registers not normalized, just as the way it is.

(2) Replace all literals with *const*. For example, *sub rsp, 07h* becomes *sub rsp, const*.

(3) Retain operations involving +, -, and * in operands.

(4) Replace the operand following the *call* operation with *foo*. For example, *call sub_401E85* becomes *call foo*.

(5) Preserve offset values in operands.

(6) Replace occurrences of *var* or *arg* in operands with *ptr*, since they are typically associated with pointers. For example, *test[ebp+arg_0],1* becomes *test [ebp+ptr], const*.

(7) Represent all the other symbols apart from the above six situations as *tag*.

## Embedding network

Taking the output of the Pre-Processing phase as input, the Embedding Network component aims to produce comprehensive function feature vectors that integrate both semantic and structural information of the disassembled functions. As illustrated in Fig. 1 and mentioned earlier, this component contains four neural networks that are designed and arranged in a specific order. Here we regard each neural network as a stage of learning in this component.

## Block embedding

The basic block embedding stage involves learning the semantic relationships between assembly instructions and converting the assembly instructions within a basic block into fixed-dimension vector representations. MSSA employs the CBOW (Continuous Bag of Words) pre-training model from Word2Vec (*Mikolov et al., 2013*) to learn the semantic relationships between assembly instructions in a self-supervised manner.

Assuming that a function $F$ consists of $n$ basic blocks, each block containing at most $m$ instruction sequences while each instruction is mapped to a $d$-dimensional vector through the CBOW model, the function $F$ can be defined as:

$$F = \{b_1, b_2, \ldots, b_i, \ldots, b_n\}, b_i \in \mathbb{R}^{m \times d}, \tag{2}$$

where $b_i$ is the feature vector of the basic block.

*Intra-block learning*

The execution behavior of assembly instructions is influenced by preceding instructions and can affect subsequent ones, creating complex dependencies. This challenges unidirectional models in capturing the complete semantic information. To address this challenge, the Intra-Block Learning stage employs Bidirectional Gated Recurrent Unit (Bi-GRU) (*Schuster & Paliwal, 1997*) to further learn the semantic relationships between assembly instructions within a single basic block. Bi-GRU processes data in both directions (forward and

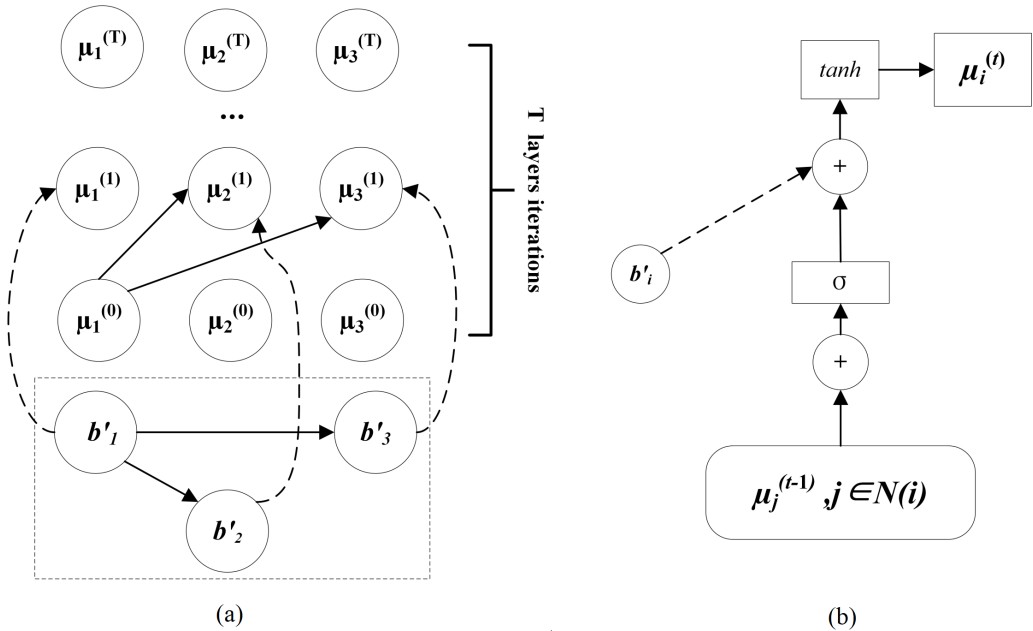

**Figure 2** Iterative network and computation process (a) Iterative network; (b) The computation process of basic block embedding vectors during a t-layer iteration.

backward), thereby capturing the semantic flow from front to back and the dependencies from back to front simultaneously. This bidirectional processing enables a comprehensive understanding of the sequence of assembly instructions within a basic block, ensuring that the semantic context is fully represented.

The feature vector of a basic block $b_i$ as derived in the Block Embedding stage can then be transformed to $b_i'$ after Intra-Block Learning. $b_i'$ can be formulated as:

$$b_i' = Bi\_GRU(b_i), b_i' \in \mathbb{R}^{m \times k}, \tag{3}$$

where $k$ is the number of neurons in the Bi-GRU and $m$ is the maximum number of instruction sequences contained in a basic block.

### Inter-block learning

As a function typically consists of multiple basic blocks, understanding the structural relationships between these basic blocks is essential for comprehensively grasping the function's overall functionality. The Inter-Block Learning stage aims to thoroughly capture this structural information and integrate it with the semantic information of each basic block as derived from the Intra-Block Learning stage. MSSA employs an iterative network to uncover and integrate structural information among basic blocks. The iterative network processes the adjacency matrix A, as derived in the Pre-Processing phase, through multiple rounds of iteration, enhancing the model's ability to generate a more robust and nuanced

representation of the function's overall structure and behavior. The iterative network is shown in Fig. 2.

Figure 2A is composed of vertices and edges, where vertices represent basic blocks, and edges represent control dependencies between basic blocks. The iterative network includes $T$ iterations. After $T$ iterations, the initial features of basic blocks are further enriched with both structural and semantic information across all basic blocks. In contrast to Gemini (*Xu et al., 2017a*), we omitted the addition operation in the output of the iterative network. We contend that simply adding the basic block features to derive function-level features risks losing critical information.

Figure 2B illustrates the computation process for the $i$-th basic block in the $t$-th iteration where the block's initial feature vector $b_i'$ is transformed into its inter-block vector $\mu_i^{(t)}$. The inputs for this process are: $b_i'$, the feature vector of basic block $i$; and the sum of $\mu_j^{(t-1)}$, namely the inter-block vectors of all the basic blocks adjacent to the $i$-th block in the adjacency matrix $A$. The computation process can be formulated as:

$$\mu_i^{(t)} = tanh(b_i' + \sigma(\sum_{j \in N(i)} \mu_j^{(t-1)})), \tag{4}$$

where the *tanh* function is used as the activation function, and $\sigma$ is a multi-layer fully connected network responsible for computing an embedding vector with more powerful representation capabilities. $N(i)$ represents the set of basic blocks adjacent to the basic block $i$. $\sigma$ can be formulated as:

$$\sigma(x) = P_1 \times ReLu(P_2 \times \ldots ReLU(P_u \times x)), \tag{5}$$

where $u$ represents the embedding depth of each basic block vertex, $x$ represents the input, and $P$ represents a fully connected parameter matrix. Through $T$ layers of iteration, the features of each vertex are propagated to other vertices *via* the iterative network, ensuring that control flow information is disseminated among all basic blocks.

Define the output layer dimension of the iterative network as $l$. The iterative network can convert the feature vectors of all basic blocks into the corresponding function's feature vector $f$. $f$ can be formulated as:

$$f = IterNet((b_1', b_2', \ldots, b_i', \ldots, b_n'), A), f \in \mathbb{R}^{n \times m \times l}. \tag{6}$$

### Function-Level learning

The high-level semantic meaning of a function extends beyond its constituent basic blocks and is implied throughout the entire structure of the function. The arrangement of all the basic blocks within a function directly reflects the core semantics and logical structure of the function, which are crucial for understanding the function's overall behavior. Therefore, conducting in-depth learning at the function level is necessary to capture these comprehensive semantics.

MSSA utilizes Bi-GRU with an attention mechanism (*Bahdanau, Cho & Bengio, 2014*), where Bi-GRU performs bidirectional learning on the function's feature vector $f$, and

the attention mechanism enhances the model's ability to focus on key features. This combination enables the model to more accurately identify the critical features that determine the function's semantics, thereby achieving a deeper semantic capture of the entire function.

The calculation formula for the attention ($att$) mechanism is as:

$$att = sum(x * softmax(tanh(x \cdot W + b))), \tag{7}$$

where $x$ represents the input, $W$ represents the weight matrix, and $b$ represents the bias vector.

Let us define the output layer dimension of Bi-GRU as $h$. The function's feature vector $f$ is transformed to $f'$. $f'$ can be formulated as:

$$f' = Bi\_GRU_{att}(f), f' \in \mathbb{R}^h. \tag{8}$$

## Similarity detection

The Similarity Detection phase is actually an application of the previous two phases and employs a siamese neural network architecture (*Feng et al., 2020*) for BCSD. Siamese networks can share parameters when processing comparative tasks and is widely used in BCSD.

We input two disassembled functions into Pre-Processing and followed by the Embedding Network component, resulting in two feature vectors respectively. We then concatenate these feature vectors, and pass them through MLP to achieve feature dimensionality reduction and enhanced classification. Finally, the output similarity score is computed through a sigmoid activation function. This score ranges between 0 and 1, where a value closer to 1 indicates that the input function pair is more similar, and a value closer to 0 indicates less similar.

The parameters are shared between the siamese networks, and the binary cross-entropy loss function [49] is used. The *Loss* function can be formulated as:

$$Loss = -\frac{1}{N}\sum_{i=1}^{N} y_i \cdot log(p(y_i)) + (1 - y_i) \cdot log(1 - p(y_i)), \tag{9}$$

where $y$ is a binary label, either 0 or 1, and $p(y)$ is the probability that the output belongs to the $y$ label.

## EVALUATION

The evaluation aims to answer the following questions:

RQ1: How does MSSA perform in BCSD compared with other baselines? ('Performance')

RQ2: How complex and efficient is MSSA in terms of parameter size and execution speed? ('Complexity and Efficiency')

RQ3: How much does the main module contribute to the performance of MSSA? ('Ablation Study')

**Table 1  BinaryCorp-3M datasets.**

| Datasets | Projects | Binaries | Functions |
|---|---|---|---|
| BinaryCorp-3M Train | 1,612 | 8,357 | 3,126,367 |
| BinaryCorp-3M Test | 364 | 1,908 | 444,574 |

## Experimental setups

### Dataset

To evaluate MSSA in detail, we used the BinaryCorp-3M (*Wang et al., 2022*) dataset, which is a large dataset constructed by jTrans, based on the ArchLinux official repository and the Arch User Repository. Each project in these repositories is compiled for one architecture x86 and five optimization levels O0, O1, O2, O3, and Os. The statistical data of the dataset are shown in Table 1.

### Baselines

We compared MSSA to the following four baselines.

Gemini (*Xu et al., 2017a*): A graph embedding network to compute the embedding vector for each node in the ACFG, ultimately forming the embedding vector for the entire ACFG (https://github.com/Yunlongs/Gemini)

Asm2Vec (*Ding, Fung & Charland, 2019*): An assembly language embedding model based on the Distributed Memory Model of Paragraph Vectors (PV-DM) (*Luo et al., 2023*) model (https://github.com/oalieno/asm2vec-pytorch).

SAFE (*Massarelli et al., 2019a*): An attention-based assembly language embedding model that uses an RNN architecture and attention mechanism to generate function embeddings (https://github.com/gadiluna/SAFE).

jTrans (*Wang et al., 2022*): A jump-aware Transformer-based model for assembly language embedding, stands as one of the state-of-the-art approaches (https://github.com/vul337/jTrans).

### Evaluation metrics

Determining whether a function pair is similar or not is a classification task. Therefore, we adopt the Accuracy, Precision, Recall, F1-Score (F1) and Area Under the Curve (AUC) metrics to evaluate the classification performance of BCSD. AUC is the area under the Receiver Operating Characteristic (ROC) curve, where the ROC curve is plotted based on the True Positive Rate (TPR) and False Positive Rate (FPR). The calculation formulas for each indicator can be formulated as:

$$Accuracy = \frac{TP + TN}{TP + TN + FP + FN}, \tag{10}$$

$$Precision = \frac{TP}{TP + FP}, \tag{11}$$

$$Recall = \frac{TP}{TP + FN}, \tag{12}$$

$$F1 = \frac{2 * Precision * Recall}{Precision + Recall}, \tag{13}$$

$$TPR = \frac{TP}{TP + FN}, \tag{14}$$

$$FPR = \frac{FP}{FP + TN}, \tag{15}$$

where the True Positive (TP) refers to a correctly defined patient sample, True Negative (TN) refers to a correctly defined healthy case, False Positive (FP) refers to a patient who has been incorrectly identified, False Negative (FN) refers to a health case with an incorrect definition.

Besides, Recall@1 and Mean Reciprocal Rank (MRR) are often used to evaluate the retrieval performance of BCSD (*Wang et al., 2022*; *Gu, Shu & Hu, 2022*; *Bottou, 2010*). Rank@1 means whether the true similar function pair has the highest score, namely ranking first, within a given set of functions. MRR calculates the average of the multiplicative inverse of the rank at which the first correct function is retrieved for a given set of functions. The calculation formulas for MRR and Recall@1 indicators can be formulated as:

$$\text{Recall@1} = \frac{1}{|F|} \sum_{f_i \in F} \tau(\text{Rank}_{f_i^{gt}} <= 1), \tag{16}$$

$$\text{MRR} = \frac{1}{|F|} \sum_{f_i \in F} \frac{1}{\text{Rank}_{f_i^{gt}}}, \tag{17}$$

where $F = f_1, f_2, f_3, \dots f_n$ represents the function pool, $f_i^{gt}$ represents the similar functions of the corresponding function $f_i$, $\text{Rank}_{f_i^{gt}}$ represents its ranking in the retrieval list, $\tau(\cdot)$ is defined as :

$$\tau(x) = \begin{cases} 0, x = \text{False} \\ 1, x = \text{True} \end{cases}. \tag{18}$$

### Hyperparameter selection

We used Adaptive Moment Estimation (Adam) (*Kingma & Ba, 2014*), Stochastic Gradient Descent (SGD) (*Dozat, 2016*), and Nesterov-accelerated Adaptive Moment Estimation (Nadam) (*Hinton & Salakhutdinov, 2006*) for optimization. In addition, we conducted multiple experiments on the number of neurons in each network based on empirical values to select the parameters with the best performance.

We ultimately chose the following hyperparameters for MSSA: the word vector length $d = 100$, the number of basic blocks $n = 20$, the number of instructions per basic block $m = 20$, the number of Bi-GRU neurons $k = 128$, the number of iterations of the network $T = 5$, the embedding depth of the iterative network $u = 3$, the output dimension of the iterative network $l = 128$, the function embedding dimension $h = 256$, and the training batch size is 1,024.

**Table 2  BCSD results on classification performance.**

| Model | Accuracy | Precision | Recall | F1 | AUC |
|---|---|---|---|---|---|
| Gemini | 0.7926 | 0.7164 | 0.9693 | 0.8237 | 0.9490 |
| Asm2Vec | 0.6916 | 0.7401 | 0.8944 | 0.7801 | 0.8411 |
| SAFE | 0.9758 | 0.9617 | 0.9882 | 0.9761 | 0.9937 |
| jTrans | 0.9708 | **0.9825** | 0.9586 | 0.9704 | 0.9941 |
| MSSA (ours) | **0.9776** | 0.9667 | **0.9894** | **0.9779** | **0.9976** |

**Notes.**
*The best results are shown in bold. The second best results are underlined.

## Performance

We divide our experiments into two parts. In the first part, we evaluate the model's classification performance through measuring the above mentioned five metrics. The second part assesses the model's retrieval performance to analyze its effectiveness in finding similar functions within a function pool.

### Classification performance

We conducted experiments to evaluate the classification performance of MSSA as compared to other four baselines using the BinaryCorp-3M dataset.

The results, presented in Table 2, show that MSSA consistently outperforms all the baselines in terms of Accuracy, Recall, F1 and AUC on classification performance, and ranks second only to jTrans regarding Precision. Specifically, MSSA outperforms jTrans by 0.0035 in AUC, surpasses SAFE by 0.0012 in Recall, 0.0018 in Accuracy, and 0.0018 in F1. However, its Precision is only 0.0158 lower than jTrans.

### Retrieval performance

We also conducted experiments to evaluate the retrieval performance of MSSA as compared to other baselines using the BinaryCorp-3M dataset. We set the function pool size to 32 and divided BinaryCorp-3M into six groups based on optimization levels: O0-O3, O0-Os, O1-O3, O1-Os, O2-O3, and O2-Os.

The BCSD results on retrieval performance are presented in Tables 3 and 4. The results indicate that the retrieval performance of MSSA is second only to jTrans. In terms of MRR, the gap between MSSA and jTrans is most significant in the O0-O3 group with MSSA being 0.1162 lower. Meanwhile, the gap is minimal in the O2-Os group with MSSA being 0.0181 lower. On average, MSSA's MRR is 0.0497 lower than jTrans. Similarly, for Recall@1, MSSA exhibits the largest gap with jTrans in the O0-O3, being 0.1772 lower. In the O2-Os group, the gap is minimal, with MSSA being 0.0343 lower. On average, MSSA's Recall@1 is 0.0811 lower than jTrans.

The results also show that the retrieval performance of MSSA surpasses other baselines. In terms of MRR, MSSA exhibits the most significant advantage over SAFE in the O2-Os group, achieving an improvement of 0.0517. In the O1-O3 group, although the performance gain of MSSA is smaller, it still achieves an increase of 0.0135. Overall, MSSA averages a 0.0273 higher MRR than SAFE. Furthermore, considering the Recall@1 metric, MSSA demonstrates the most notable difference compared to SAFE in the O2-Os group, achieving

**Table 3  BCSD results on retrieval metric MRR.**

| Models | MRR | | | | | | |
| --- | --- | --- | --- | --- | --- | --- | --- |
| | O0-O3 | O0-Os | O1-O3 | O1-Os | O2-O3 | O2-Os | Average |
| Gemini | 0.6090 | 0.6878 | 0.8076 | 0.8040 | 0.9007 | 0.8128 | 0.7703 |
| SAFE | 0.8221 | 0.8610 | 0.9177 | 0.9327 | 0.9374 | 0.9095 | 0.8967 |
| Asm2Vec | 0.3146 | 0.3220 | 0.6253 | 0.6007 | 0.7603 | 0.6491 | 0.5453 |
| jTrans | **0.9626** | **0.9591** | **0.9790** | **0.9768** | **0.9857** | **0.9793** | **0.9737** |
| MSSA (ours) | 0.8464 | 0.8887 | 0.9312 | 0.9507 | 0.9660 | 0.9612 | 0.9240 |

**Notes.**
*The best results are shown in bold. The second best results are underlined.

**Table 4  BCSD results on retrieval metric Recall@1.**

| Models | Recall@1 | | | | | | |
| --- | --- | --- | --- | --- | --- | --- | --- |
| | O0-O3 | O0-Os | O1-O3 | O1-Os | O2-O3 | O2-Os | Average |
| Gemini | 0.4463 | 0.5516 | 0.7141 | 0.7017 | 0.8473 | 0.7237 | 0.6641 |
| SAFE | 0.7346 | 0.7692 | 0.8648 | 0.8868 | 0.8967 | 0.8461 | 0.8330 |
| Asm2Vec | 0.1865 | 0.1926 | 0.5126 | 0.4838 | 0.6839 | 0.5461 | 0.4343 |
| jTrans | **0.9362** | **0.9324** | **0.9638** | **0.9592** | **0.9746** | **0.9641** | **0.9550** |
| MSSA (ours) | 0.7590 | 0.8177 | 0.8868 | 0.9109 | 0.9395 | 0.9298 | 0.8739 |

**Notes.**
*The best results are shown in bold. The second best results are underlined.

a boost of 0.0837. While the advantage of MSSA in the O1-O3 group is relatively modest, it still manages to achieve an increase of 0.022. On average, MSSA outperforms SAFE by 0.0409 in the Recall@1.

## Complexity and efficiency

We conducted experiments to evaluate the complexity and efficiency of MSSA in contrast to other comparable approaches, namely SAFE and jTrans which have better performance than Gemini and Asm2Vec as indicated in the previous experiment, using the BinaryCorp-3M dataset again. The complexity metric is measured by the number of model parameters, while the efficiency is evaluated based on the execution speed of a model on a CPU environment. Note that we modified MSSA and SAFE to detect one by one of a pair of functions, as same as jTrans, in this section.

The number of model parameters is counted through the corresponding library functions in the deep learning frameworks for each baseline model. MSSA uses *summary()*, SAFE employs *trainable_variables()*, and jTrans use *parameters()*.

The experiments were conducted on a Windows 10 desktop computer, equipped with an Intel Core i5-11400 CPU, 16 GB RAM, and without Graphics Processing Unit (GPU).

We run each test for 10 times, and Table 5 shows the average results. The experimental results show that MSSA has the fewest parameters in the backbone network, with SAFE closely following in second place, while jTrans has the highest number of parameters. Even after incorporating pre-trained parameters, MSSA maintains top ranking in total parameter count, which remains lower than that of SAFE and jTrans. In terms of execution

**Table 5  Results on complexity and efficiency.**

| Models | Number of model parameters | | | Execution Speed (pairs/s) |
|---|---|---|---|---|
| | Pre-trained | Backbone | Total | |
| SAFE | 52,768,300 | 4,477,000 | 57,245,300 | 6.78 |
| jTrans | 73,110,528 | 14,766,336 | 87,876,864 | 0.74 |
| MSSA (ours) | **16,109,336** | **379,229** | **16,488,565** | **32.20** |

speed, MSSA exhibited the highest performance at 32.20 pairs per second. SAFE followed closely with a speed of 6.78. As for jTrans, its execution speed is very slow, only reaching 0.74.

In summary, MSSA strikes a balance between complexity and efficiency, maintaining a low parameter count while delivering high performance. Additionally, its ability to be deployed in a CPU environment further enhances its versatility for broader applications.

## Ablation study

In this section, we try to find out the key factors that affect the performance of MSSA in BCSD. We evaluate the following three parts:

(1) How does the Intra-Block Learning stage affect the performance of MSSA?
(2) How does the structural information in the Inter-Block Learning stage affect the performance of MSSA?
(3) How does the attention mechanism in the Function-Level Learning stage affect the performance of MSSA?

The results are presented in Fig. 3. The MSSA label in the figure represents the complete approach, MSSA-NIB represents the approach without the Intra-block stage, MSSA-NAM represents the approach without the Inter-block stage, and MSSA-NA represents the approach without the attention mechanism. In comparing MSSA and MSSA-NIB, we found that learning the sequence of assembly instructions within the basic blocks is essential. The inclusion of the Intra-block stage significantly enhances MSSA's performance. Specifically, its introduction improves MSSA's MRR metric by 30.7% and Recall@1 by 52.5% compared to the absence of this stage.

In comparing MSSA and MSSA-NAM, we determined that in understanding the semantic relationships between basic blocks, the structural information of these blocks plays a pivotal role. It enables the model to capture the logical relationships between different basic blocks more accurately, thereby enhancing MSSA's performance. Specifically, introducing the adjacency matrix of basic blocks during the Inter-Block Learning stage to leverage structural information increases MSSA's MRR metric by 0.1493, and improves Recall@1 by 0.2257 compared to not utilizing this information.

In comparing MSSA and MSSA-NA, we noted that while the inclusion of the attention mechanism in the Function-Level Learning phase enhances MSSA's performance, its impact is relatively minor. Specifically, compared to its absence, the introduction of the

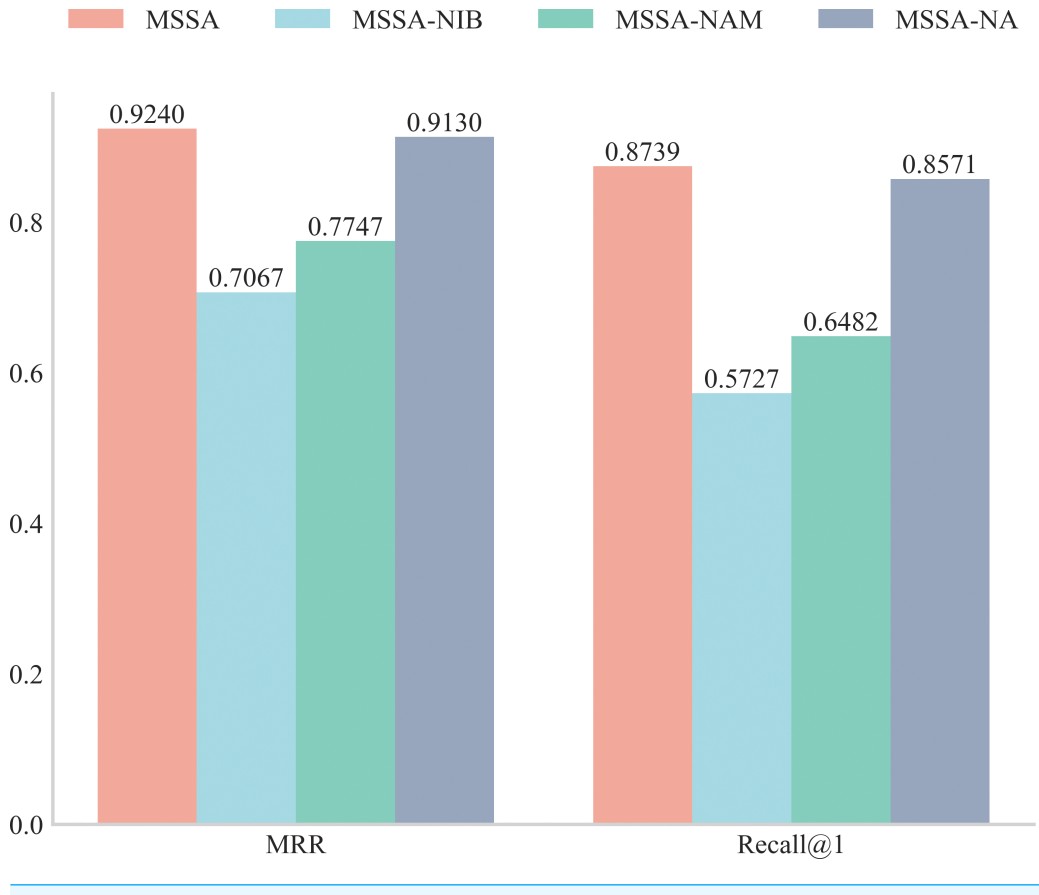

**Figure 3** **BCSD results on ablation study of MSSA.**

attention mechanism increases MSSA's MRR metric by only 0.011 and Recall@1 by only 0.017.

## DISCUSSION

In this study, we introduce MSSA, a multi-stage semantic-aware neural network designed for binary code similarity detection. MSSA effectively integrates both semantic and structural information from assembly instructions through four stages of learning: Block Embedding, Intra-Block Learning, Inter-Block Learning, and Function-Level Learning. This approach enables MSSA to achieve a deeper understanding of binary code semantics.

Our experimental results demonstrate that MSSA outperforms existing methods like Gemini, Asm2Vec, SAFE, and jTrans in classification tasks, while ranking just behind the Transformer-based jTrans in retrieval performance. Notably, MSSA's backbone network consists of only 0.38 million parameters, significantly smaller than Transformer-based models, making it highly advantageous in resource-constrained environments, especially those lacking GPUs. Specifically, the results of the ablation study demonstrate that the Intra-Block and Inter-Block Learning stages contribute most significantly to the performance of MSSA, indicating that they can effectively capture both semantic and structural information

of assembly instructions for code embeddings. This in turn has implications for other code tasks, such as code generation, which may also benefit from gradual learning at varying levels of granularity.

However, there is still room for improvement in MSSA's retrieval metrics, an area where Transformer-based models excel. Additionally, MSSA faces challenges in detecting obfuscated binary code. While Transformer models excel in retrieval performance, their large number of parameters and high training costs remain significant drawbacks. Future research will focus on optimizing and balancing four key factors: model size, training cost, detection speed, and overall performance. Moreover, obfuscation techniques introduce extraneous code that disrupts original structures and semantics, presenting a challenge for MSSA in learning and detecting such patterns effectively. Future work will aim to enhance the model's robustness against obfuscation, potentially through the incorporation of anti-obfuscation techniques or by improving generalization to obfuscated patterns.

In conclusion, MSSA offers a lightweight yet powerful solution for binary code similarity detection, balancing high performance with resource efficiency. This opens up new possibilities for deploying advanced BCSD in resource-constrained settings and has significant implications for both research and practical applications in related fields.

## CONCLUSIONS

In this study, we have proposed MSSA, a multi-stage semantic-aware neural network for BCSD. MSSA efficiently integrates semantics and structural information of assembly instructions at multiple levels—within basic blocks, between basic blocks, and across entire function—facilitating thorough understanding and precise similarity detection of binary code. Our experimental results demonstrate MSSA's superior classification performance compared to established approaches such as Gemini, Asm2Vec, SAFE and jTrans. While MSSA's retrieval performance slightly trails behind the Transformer-based jTrans, its backbone network parameter size of only 0.38M positions MSSA as an optimal choice for deployment in CPU environments. MSSA represents a new lightweight approach for BCSD, offering novel insights and tools that can impact research and applications across related fields.

### Funding

This paper was funded by the Chongqing Technology Innovation and Application Development Special Major Project under Grant No. CSTB2023TIAD-STX0034. The funders had no role in study design, data collection and analysis, decision to publish, or preparation of the manuscript.

### Grant Disclosures

The following grant information was disclosed by the authors:
Chongqing Technology Innovation and Application Development Special Major Project under: CSTB2023TIAD-STX0034.

## Competing Interests

The authors declare there are no competing interests.

## Author Contributions

- Bangrui Wan conceived and designed the experiments, analyzed the data, performed the computation work, authored or reviewed drafts of the article, and approved the final draft.
- Jianjun Zhou conceived and designed the experiments, performed the experiments, performed the computation work, prepared figures and/or tables, authored or reviewed drafts of the article, and approved the final draft.
- Ying Wang performed the experiments, performed the computation work, prepared figures and/or tables, and approved the final draft.
- Feng Chen analyzed the data, authored or reviewed drafts of the article, and approved the final draft.
- Ying Qian conceived and designed the experiments, authored or reviewed drafts of the article, and approved the final draft.

## Data Availability

The code is available at Github and Zenodo:

- https://github.com/SQAbin/MSSA.

- SQA. (2024). SQAbin/MSSA: MSSA (BCSD). Zenodo. https://doi.org/10.5281/zenodo.13881709.

The third party dataset from the approach jTrans is available at Github: https://github.com/vul337/jTrans.

## Supplemental Information

Supplemental information for this article can be found online at http://dx.doi.org/10.7717/peerj-cs.2504#supplemental-information.

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
