# Peer review of "MSSA: multi-stage semantic-aware neural network for binary code similarity detection"

_PeerJ Computer Science, doi:10.7717/peerj-cs.2504_

## Round 0.1 · original submission · Major Revisions

Dear authors,

Thank you for the submission. The reviewers’ comments are now available. It is not suggested that your article be published in its current format. We do, however, advise you to revise the paper in light of the reviewers’ comments and concerns before resubmitting it. The followings should also be addressed:

1. Many of the equations are part of the related sentences. Attention is needed for correct sentence formation.
2. Equations should be used with correct equation number. Please do not use “as follows”, “given as”, etc. Explanation of the equations should also be checked. All variables should be written in italic as in the equations. Their definitions and boundaries should be defined. Necessary references should be provided.
3. All of the values for the parameters of all algorithms selected for comparison should be given.
4. Pay special attention to the usage of abbreviations. Spell out the full term at its first mention, indicate its abbreviation in parenthesis and use the abbreviation from then on.
5. Some more recommendations and conclusions should be discussed about the paper considering the experimental results. The conclusion section is weak. There is also no discussion section about the results. It should briefly describe the results of the study and some more directions for further research. You should describe the academic implications, main findings, shortcomings and directions for future research in the conclusion section. The conclusion in its current form is generally confused. What will be happen next? What we supposed to expect from the future papers? So rewrite it and consider the following comments:
- Highlight your analysis and reflect only the important points for the whole paper.
- Mention the benefits
- Mention the implication in the last of this section.
6. Reviewers have advised you to provide specific references. You are welcome to add them if you think they are relevant and useful . However, you are under no obligation to include them, and if you do not, it will not affect my decision.

Best wishes,

Reviewer 1 ·

Basic reporting

1. The mathematical analysis of the proposed scheme is very weak.
2. Authors should argue their choice of performance evaluation indicators.
3. Performing a comparison-based analysis with a recent and related approach under equal conditions could help to improve and justify the contribution of the paper.
4. More clarifications and highlights about the research gaps in the related works section need to be included.
5. Please extend the results by adding new experiments.

Experimental design

no comment

Validity of the findings

no comment

Additional comments

no comment

Reviewer 2 ·

Basic reporting

Very well.

Experimental design

Good.

Validity of the findings

The authors obtained important and valid findings.

Additional comments

In this study, the authors propose a multi-stage semantic aware neural network (MSSA) for Binary Code Similarity Detection (BCSD). I think that the study can be published by making some corrections below:
1. The mathematical formulas of the evaluation metrics (Accuracy, Precision, Recall, F1, AUC, ROC Curve, True Positive Rate (TPR) and False Positive Rate (FPR)) should be given. Support can be obtained from https://doi.org/10.1016/j.apacoust.2023.109476 and https://doi.org/10.1016/j.compbiomed.2023.107153 studies.

2. In the class study (first application), it is recommended to provide the ROC Curve and Confusion Matrix results of the proposed model.
3. Did you use any optimization techniques when determining the Hyperparameters of the models?
4. Isn’t the tracking batch size set to 1024 too high? Doesn’t this value negatively affect the performance of the model?
5. MRR and Recall@1 metrics should be given mathematically.
6. When the results of Table 3 and Table 4 are examined in the second application (Retrieval Performance), a performance difference of 12-18% is seen for the proposed method according to O0-O3 compared to the jTrans technique. This is a very big difference. It is recommended that the method you proposed be improved in this application.

Annotated reviews are not available for download in order to protect the identity of reviewers who chose to remain anonymous.

Reviewer 3 ·

Basic reporting

Clarity and Language: The manuscript is written in clear and professional English, with precise language used throughout. The concepts are well-explained, and technical jargon is appropriately utilized, making it accessible to the target audience. However, a few minor grammatical errors and awkward phrasings could be polished for smoother reading.

Introduction and Background: The introduction provides a comprehensive overview of the topic, emphasizing the importance of Binary Code Similarity Detection (BCSD) and setting the context for the research. The background section is thorough, covering various traditional and modern approaches to BCSD, including Transformer-based methods. The authors successfully identify gaps in the existing research, justifying the need for the proposed MSSA model.

Literature Review: The manuscript is well-referenced, citing relevant and up-to-date studies in the field. The authors have included a variety of sources, from foundational works to recent advancements, ensuring a robust theoretical foundation.

Structure and Figures: The structure of the paper conforms to standard academic norms. The sections are logically organized, and the flow from one section to the next is seamless. Figures and tables are relevant, well-labeled, and of high quality. They effectively illustrate the concepts and results discussed in the text.

Data Availability: The raw data and code associated with the research are provided, adhering to PeerJ’s data-sharing policies. This transparency enhances the reproducibility of the study.

Experimental design

Research Question: The research question is well-defined, relevant, and timely. The study focuses on improving the efficiency and effectiveness of BCSD models, particularly in resource-constrained environments. The authors clearly state how their work addresses a significant gap in the field.

Methodology: The experimental design is rigorous and well-executed. The authors describe the proposed MSSA model in detail, including its architecture, the specific techniques used at different stages, and how these techniques integrate semantic and structural information. The description is comprehensive enough to allow replication of the study by other researchers.

Dataset and Baselines: The choice of the BinaryCorp-3M dataset is appropriate, given its size and relevance to the BCSD task. The selection of baseline models (Gemini, Asm2Vec, SAFE, and jTrans) is also appropriate, as they represent a broad spectrum of approaches in the field. The comparisons made against these baselines are well-justified and provide a solid basis for evaluating the performance of MSSA.

Evaluation Metrics: The authors employ a comprehensive set of evaluation metrics (Accuracy, Precision, Recall, F1, AUC, MRR, and Recall@1) to assess the performance of their model. This multi-faceted approach ensures that the model’s effectiveness is evaluated from various perspectives, increasing the validity of the findings.

Validity of the findings

Results: The findings are well-supported by the data. The experimental results demonstrate that MSSA outperforms several established models in classification performance and is second only to a Transformer-based model in retrieval performance. The improvements are statistically significant, and the authors provide detailed explanations for the observed performance gains.

Interpretation: The interpretation of the results is sound and aligns with the research objectives. The authors carefully analyze the contribution of different components of the MSSA model through an ablation study, providing insights into why their model performs better than the baselines.

Limitations: The manuscript discusses the limitations of the study, particularly the challenges MSSA faces in detecting obfuscated binary code. This acknowledgment of limitations adds to the credibility of the research and suggests areas for future work.

Additional comments

Strengths:
The paper addresses a significant problem in BCSD with a novel approach that balances performance and computational efficiency.
The detailed explanation of the MSSA architecture and its components is a strong point of the paper.
The comprehensive experimental evaluation and comparison with state-of-the-art methods enhance the paper’s contribution to the field.

Suggestions for Improvement:
The language could be polished in some sections for clarity and readability.
While the figures are generally well-done, adding more detailed captions could help in making them more self-explanatory.
A more in-depth discussion of potential applications of MSSA in real-world scenarios could be beneficial.
And several recent works about efficient deep learning [1-4] are suggested to be discussed to further extend the application on resource-constrained inference scenarios of the proposed method.
[1] Distribution-sensitive Information Retention for Accurate Binary Neural Network. IJCV 2023
[2] Diverse Sample Generation: Pushing the Limit of Generative Data-free Quantization. TPAMI 2023
[3] BiBench: Benchmarking and Analyzing Network Binarization. ICML 2023
[4] Accurate LoRA-Finetuning Quantization of LLMs via Information Retention. ICML 2024

Overall, the manuscript presents a well-conducted study with significant contributions to the field of BCSD. The findings are robust and supported by comprehensive experimental evidence, making this paper a valuable addition to the existing literature.

---

## Round 0.2 · accepted · Accept

Dear Authors,

Thank you for the revision. One of the original reviewers did not respond to the invitation for the revised paper. Other two reviewers accept the paper and I also confirm that your paper is improved and can be accepted for publication.

Best wishes,

Reviewer 1 ·

Basic reporting

According to the response letter, the paper has been revised, and the current version of the manuscript is acceptable for publication.

Experimental design

NA

Validity of the findings

NA

Additional comments

NA

Reviewer 3 ·

Basic reporting

no comment

Experimental design

no comment

Validity of the findings

no comment

Additional comments

no comment